# Anti-efficient encoding in emergent communication

Rahma Chaabouni[1,2], Eugene Kharitonov[1], Emmanuel Dupoux[1,2] and Marco Baroni[1,3]

[1]*Facebook AI Research*
[2]*Cognitive Machine Learning (ENS - EHESS - PSL Research University - CNRS - INRIA)*
[3]*ICREA*
{rchaabouni,kharitonov,dpx,mbaroni}@fb.com

## Abstract

Despite renewed interest in emergent language simulations with neural networks, little is known about the basic properties of the induced code, and how they compare to human language. One fundamental characteristic of the latter, known as Zipf's Law of Abbreviation (ZLA), is that more frequent words are efficiently associated to shorter strings. We study whether the same pattern emerges when two neural networks, a "speaker" and a "listener", are trained to play a signaling game. Surprisingly, we find that networks develop an *anti-efficient* encoding scheme, in which the most frequent inputs are associated to the longest messages, and messages in general are skewed towards the maximum length threshold. This anti-efficient code appears easier to discriminate for the listener, and, unlike in human communication, the speaker does not impose a contrasting least-effort pressure towards brevity. Indeed, when the cost function includes a penalty for longer messages, the resulting message distribution starts respecting ZLA. Our analysis stresses the importance of studying the basic features of emergent communication in a highly controlled setup, to ensure the latter will not depart too far from human language. Moreover, we present a concrete illustration of how different functional pressures can lead to successful communication codes that lack basic properties of human language, thus highlighting the role such pressures play in the latter.

## 1 Introduction

There is renewed interest in simulating language emergence among neural networks that interact to solve a task, motivated by the desire to develop automated agents that can communicate with humans [e.g., Havrylov and Titov, 2017, Lazaridou et al., 2017, 2018, Lee et al., 2018]. As part of this trend, several recent studies analyze the properties of the emergent codes [e.g., Kottur et al., 2017, Bouchacourt and Baroni, 2018, Evtimova et al., 2018, Lowe et al., 2019, Graesser et al., 2019]. However, these analyses generally consider relatively complex setups, when very basic characteristics of the emergent codes have yet to be understood. We focus here on one such characteristic, namely the length distribution of the messages that two neural networks playing a simple signaling game come to associate to their inputs, in function of input frequency.

In his pioneering studies of lexical statistics, George Kingsley Zipf noticed a robust trend in human language that came to be known as Zipf's Law of Abbreviation (ZLA): There is an inverse (non-linear) correlation between word frequency and length [Zipf, 1949, Teahan et al., 2000, Sigurd et al., 2004, Strauss et al., 2007]. Assuming that shorter words are easier to produce, this is an efficient encoding strategy, particularly effective given Zipf's other important discovery that word distributions are highly skewed, following a power-law distribution. Indeed, in this way language approaches an optimal code in information-theoretic terms [Cover and Thomas, 2006]. Zipf, and many after him, have thus used ZLA as evidence that language is shaped by functional pressures toward effort

minimization [e.g., Piantadosi et al., 2011, Mahowald et al., 2018, Gibson et al., 2019]. However, others [e.g., Mandelbrot, 1954, Miller et al., 1957, Ferrer i Cancho and del Prado Martín, 2011, del Prado Martín, 2013] noted that some random-typing distributions also respect ZLA, casting doubts on functional explanations of the observed pattern.

We study a *Speaker* network that gets one out of $1K$ distinct one-hot vectors as input, randomly drawn from a power-law distribution (so that frequencies are extremely skewed, like in natural language). Speaker transmits a variable-length *message* to a *Listener* network. Listener outputs a one-hot vector, and the networks are rewarded if the latter is identical to the input. There is no direct supervision on the message, so that the networks are free to create their own "language". The networks develop a successful communication system that does *not* exhibit ZLA, and is indeed *anti-efficient*, in the sense that all messages are long, and the most frequent inputs are associated to the longest messages. Interestingly, a similar effect is observed in artificial human communication experiments, in conditions in which longer messages do not demand extra effort to speakers, so that they are preferred as they ease the listener discrimination task [Kanwal et al., 2017]. Our Speaker network, unlike humans, has no physiological pressure towards brevity [Chaabouni et al., 2019], and our Listener network displays an *a priori* preference for longer messages. Indeed, when we penalize Speaker for producing longer strings, the emergent code starts obeying ZLA. We examine the implications of our findings in the Discussion.

## 2 Setup

### 2.1 The game

We designed a variant of the Lewis signaling game [Lewis, 1969] in which the input distribution follows a power-law distribution. We think of these inputs as a vocabulary of distinct abstract *word types*, to which the agents will assign specific word forms while learning to play the game. We leave it to further research to explore setups in which word type and form distributions co-evolve [Ferrer i Cancho and Díaz-Guilera, 2007]. Formally, the game proceeds as follows:

1. The Speaker network receives one of $1K$ distinct one-hot vectors as input $i$. Inputs are not drawn uniformly, but, like in natural language, from a power-law distribution. That is, the $r^{th}$ most frequent input $i_r$ has probability $\frac{1}{r \times \sum_{k=1}^{1000} \frac{1}{k}}$ to be sampled, with $r \in [\![1, ..., 1000]\!]$. Consequently, the probability of sampling the $1^{st}$ input is 0.13 while the probability of sampling the $1000^{th}$ one is 1000 times lower.

2. Speaker chooses a sequence of symbols from its alphabet $A = \{s_1, s_2..., s_{a-1}, \texttt{eos}\}$ of size $|A| = a$ to construct a message $m$, terminated as soon as Speaker produces the 'end-of-sequence' token $\texttt{eos}$. If Speaker has not yet emitted $\texttt{eos}$ at $\texttt{max\_len} - 1$, it is stopped and $\texttt{eos}$ is appended at the end of its message (so that all messages are suffixed with $\texttt{eos}$ and no message is longer than $\texttt{max\_len}$).

3. The Listener network consumes $m$ and outputs $\hat{i}$.

4. The agents are successful if $i = \hat{i}$, that is, Listener reconstructed Speaker's input.

The game is implemented using the EGG toolkit [Kharitonov et al., 2019], and the code can be found at `https://github.com/facebookresearch/EGG/tree/master/egg/zoo/channel`.

### 2.2 Architectures

As standard in current emergent-language simulations [e.g., Lazaridou et al., 2018], both agents are implemented as single-layer LSTMs [Hochreiter and Schmidhuber, 1997]. Speaker's input is a $1K$-dimensional one-hot vector $i$, and the output is a sequence of symbols, defining message $m$. This sequence is generated as follows. A linear layer maps the input vector into the initial hidden state of Speaker's LSTM cell. Next, a special start-of-sequence symbol is fed to the cell. At each step of the sequence, the output layer defines a Categorical distribution over the alphabet. At training time, we sample from this distribution. During evaluation, we select the symbol greedily. Each selected symbol is fed back to the LSTM cell. The dimensionalities of the hidden state vectors are part of the hyper-parameters we explore (Appendix A.1). Finally, we initialize the weight matrices of our agents

with a uniform distribution with support in $[-\frac{1}{\sqrt{\texttt{input\_size}}}, \frac{1}{\sqrt{\texttt{input\_size}}}]$, where `input_size` is the dimensionality of the matrix input (Pytorch default initialization).

Listener consumes the entire message $m$, including `eos`. After `eos` is received, Listener's hidden state is passed through a fully-connected layer with softmax activation, determining a Categorical distribution over $1K$ indices. This distribution is used to calculate the cross-entropy loss w.r.t. the ground-truth input, $i$.

The joint Speaker-Listener architecture can be seen as a discrete auto-encoder [Liou et al., 2014].

## 2.3 Optimization

The architecture is not directly differentiable, as messages are discrete-valued. In language emergence, two approaches are dominantly used: Gumbel-Softmax relaxation [Maddison et al., 2016, Jang et al., 2016] and REINFORCE [Williams, 1992]. We also experimented with the approach of Schulman et al. [2015], combining REINFORCE and stochastic backpropagation to estimate gradients. Preliminary experiments showed that the latter algorithm (to be reviewed next) results in the fastest and most stable convergence, and we used it in all the following experiments. However, the main results we report were also observed with the other algorithms, when successful.

We denote by $\boldsymbol{\theta}_s$ and $\boldsymbol{\theta}_l$ the Speaker and Listener parameters, respectively. $\mathcal{L}$ is the cross-entropy loss, that takes the ground-truth one-hot vector $i$ and Listener's output $L(m)$ distribution as inputs. We want to minimize the expectation of the cross-entropy loss $\mathbb{E}\,\mathcal{L}(i, L(m))$, where the expectation is calculated w.r.t. the joint distribution of inputs and message sequences. The gradient of the following surrogate function is an unbiased estimate of the gradient $\nabla_{\boldsymbol{\theta}_s \cup \boldsymbol{\theta}_l} \mathbb{E}\,\mathcal{L}(i, L(m))$:

$$\mathbb{E}\left[\mathcal{L}(i, L(m; \boldsymbol{\theta}_l)) + (\{\mathcal{L}(i, L(m; \boldsymbol{\theta}_l)\} - b) \log P_s(m|\boldsymbol{\theta}_s)\right] \tag{1}$$

where $\{\cdot\}$ is the stop-gradient operation, $P_s(m|\boldsymbol{\theta}_s)$ is the probability of producing the sequence $m$ when Speaker is parameterized with vector $\boldsymbol{\theta}_s$, and $b$ is a running-mean baseline used to reduce the estimate variance without introducing a bias. To encourage exploration, we also apply an entropy regularization term [Williams and Peng, 1991] on the output distribution of the speaker agent.

Effectively, under Eq. 1, the gradient of the loss w.r.t. the Listener parameters is found via conventional backpropagation (the first term in Eq. 1), while Speaker's gradient is found with a REINFORCE-like procedure (the second term). Once the gradient estimate is obtained, we feed it into the Adam [Kingma and Ba, 2014] optimizer. We explore different learning rate and entropy regularization coefficient values (Appendix A.1).

We train agents for 2500 episodes, each consisting of 100 mini-batches, in turn including 5120 inputs sampled from the power-law distribution with replacement. After training, we present to the system each input once, to compute accuracy by giving equal weight to all inputs, independently of amount of training exposure.

## 2.4 Reference distributions

As ZLA is typically only informally defined, we introduce 3 reference distributions that display efficient encoding and arguably respect ZLA.

### 2.4.1 Optimal code

Based on standard coding theory [Cover and Thomas, 2006], we design an *optimal code* (OC) guaranteeing the shortest average message length given a certain alphabet size and the constraint that all messages must end with `eos`. The *shortest* messages are deterministically associated to the *most frequent* inputs, leaving longer ones for less frequent ones. The length of the message associated to an input is determined as follows. Let $A = \{s_1, s_2...s_{a-1}, \texttt{eos}\}$ be the alphabet of size $a$ and $i_r$ be the $r^{th}$ input when ranked by frequency. Then $i_r$ is mapped to a message of length

$$l_{i_r} = min\{n : \sum_{k=1}^{n} (a-1)^{k-1} \geq r\} \tag{2}$$

For instance, if $a = 3$, then there is only one message of length 1 (associated to the most frequent referent), 2 of length 2, 4 of length 3 etc.[1] Section 2 of Ferrer i Cancho et al. [2013] presents a proof of how this encoding is the maximally efficient one.

### 2.4.2 Monkey typing

Natural languages respect ZLA without being as efficient as OC. It has been observed that *Monkey typing* (MT) processes, whereby a monkey hits random typewriter keys including a space character, produce word length distributions remarkably similar to those attested in natural languages [Simon, 1955, Miller et al., 1957]. We thus adapt a MT process to our setup, as a less strict benchmark for network efficiency.[2]

We first sample an input without replacement according to the power-law distribution, then generate the message to be associated with it. We repeat the process until all inputs are assigned a unique message. The message is constructed by letting a monkey hit the $a$ keys of a typewriter uniformly at random ($p = 1/a$), subject to these constraints: (i) The message ends when the monkey hits `eos`. (ii) A message cannot be longer than a specified length `max_len`. If the monkey has not yet emitted `eos` at `max_len` $- 1$, it is stopped and `eos` is appended at the end of the message. (iii) If a generated message is identical to one already used, it is rejected and another is generated.

For a given length $l$, there are only $(a - 1)^{l-1}$ different messages. Moreover, for a random generator with the `max_len` constraint, the probability of generating a message of length $l$ is:

$$P_l = p \times (1 - p)^{l-1}, \text{if } l < \texttt{max\_len} \text{ and } P_{\texttt{max\_len}} = (1 - p)^{\texttt{max\_len}-1} \qquad (3)$$

From these calculations, we derive two qualitative observations about MT. First, as we fix `max_len` and increase $a$ (decrease $p = 1/a$), more generated messages will reach `max_len`. Second, when $a$ is small and `max_len` is large (as in early MT studies where `max_len` was infinite), a ZLA-like distribution emerges, due to the finite number of *different* messages of length $l$. Indeed, for any $l$ less than `max_len`, $P_l$ strictly decreases as $l$ grows. Then, for given inputs, the monkey is likely to start by generating messages of the most probable length (that is, 1). As we exhaust all unique messages of this length, the process starts generating messages of the next probable length (i.e., 2) and so on. Figure A.1 in Appendix A.2 confirms experimentally that our MT distribution respects ZLA for $a \leq 10$ and various `max_len`.

### 2.4.3 Natural language

We finally consider word length distributions in natural language corpora. We used pre-compiled English, Arabic, Russian and Spanish frequency lists from `http://corpus.leeds.ac.uk/serge/`, extracted from corpora of internet text containing between $200M$ (Russian) and $16M$ words (Arabic). For direct comparability with input set cardinality in our simulations, we only looked at the distribution of the top 1000 most frequent words, after merging lower- and upper-cased forms, and removing words containing non-alphabetical characters. The resulting word frequency distributions obeyed power laws with exponents between $-0.81$ and $-0.92$ (we used $-1$ to generate our inputs). Alphabet sizes are as follows: 30 (English), 31 (Spanish), 47 (Russian), 59 (Arabic). These are larger than normative sizes, as unfiltered Internet text will occasionally include foreign characters (e.g., accented letters in English text). Contrary to previous reference distributions, we cannot control `max_len` and alphabet size. We hence compare human and network distributions only in the adequate settings. In the main text, we present results for the languages with the smallest (English) and largest (Arabic) alphabets. The distributions of the other languages are comparable, and presented in Appendix A.3.

## 3 Experiments

### 3.1 Characterizing the emergent encoding

We experiment with alphabet sizes $a \in [3, 5, 10, 40, 1000]$. We chose mainly small alphabet sizes to minimize a potential bias in favor of long messages: For high $a$, randomly generating long messages becomes more likely, as the probability of outputting `eos` at random becomes lower. At the other

extreme, we also consider $a = 1000$, where the Speaker could in principle successfully communicate using at most 2-symbol messages (as Speaker needs to produce `eos`). Finally, $a = 40$ was chosen to be close to the alphabet size of the natural languages we study (mean alphabet size: $41.75$).

After fixing $a$, we choose `max_len` so that agents have enough capacity to describe the whole input space ($|I| = 1000$). For a given $a$ and `max_len`, Speaker cannot encode more inputs than the message space size $M_a^{\texttt{max\_len}} = \sum_{j=1}^{\texttt{max\_len}} (a-1)^{j-1}$. We experiment with $\texttt{max\_len} \in [2, 6, 11, 30]$. We couldn't use higher values because of memory limitations. Furthermore, we studied the effect of $D = \frac{M_a^{\texttt{max\_len}}}{|I|}$. While making sure that this ratio is at least 1, we experiment with low values, where Speaker would have to use nearly the whole message space to successfully denote all inputs. We also considered settings with significantly larger $D$, where constructing $1K$ distinct messages might be an easier task.

We train models for each $(\texttt{max\_len}, a)$ setting and agent hyperparameter choice (4 seeds per choice). We consider runs successful if, after training, they achieve an accuracy above $99\%$ on the full input set (i.e., less than 10 miss-classified inputs). As predicted, the higher $D$ is, the more accurate the agents become. Indeed, agents need much larger $D$ than strictly necessary in order to converge. We select for further analysis only those $(\texttt{max\_len}, a)$ choices that resulted in more than 3 successful runs (mean number of successful runs across the reported configurations is 25 out of 48). Moreover, we focus here on configurations with $\texttt{max\_len} = 30$, as the most comparable to natural language.[3] We present results for all selected configurations (confirming the same trends) in Appendix A.4.

Figure 1 shows message length distribution (averaged across all successful runs) as a function of input frequency rank, compared to our reference distributions. The MT results are averaged across 25 different runs. We show the Arabic and English distributions in the plot containing the most comparable simulation settings $(30, 40)$.

Across configurations, we observe that Speaker messages greatly depart from ZLA. There is a clear general preference for longer messages, that is strongest *for the most frequent inputs*, where Speaker outputs messages of length `max_len`. That is, in the emergent encoding, more frequent words are longer, making the system obey a sort of "*anti*-ZLA" (see Appendix A.6 for confirmation that this anti-efficient pattern is statistically significant). Consequently, the emergent language distributions are well above all reference distributions, except for MT with $a = 1000$, where the large alphabet size leads to uniformly long words, for reasons discussed in Section 2.4.2. Finally, the lack of efficiency in emergent language encodings is also observed when inputs are uniformly distributed (see Appendix A.5).

Although some animal signing systems disobey ZLA, due to specific environmental constraints [e.g., Heesen et al., 2019], a large survey of human and animal communication did not find any case of significantly *anti*-efficient systems [Ferrer i Cancho et al., 2013], making our finding particularly intriguing.

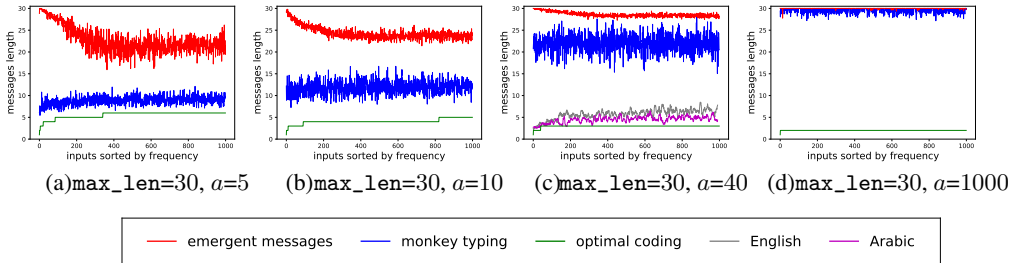

(a)`max_len`=30, $a$=5    (b)`max_len`=30, $a$=10    (c)`max_len`=30, $a$=40    (d)`max_len`=30, $a$=1000

— emergent messages   — monkey typing   — optimal coding   — English   — Arabic

Figure 1: Mean message length across successful runs as a function of input frequency rank, with reference distributions. For readability, we smooth natural language distributions by reporting the sliding average of 10 consecutive lengths.

## 3.2 Causes of anti-efficient encoding

We explore the roots of anti-efficiency by looking at the behavior of untrained Speakers and Listeners. Earlier work conjectured that ZLA emerges from the competing pressures to communicate in a perceptually distinct *and* articulatorily efficient manner [Zipf, 1949, Kanwal et al., 2017]. For our networks, there is a clear pressure from Listener in favour of ease of message discriminability, but Speaker has no obvious reason to save on "articulatory" effort. We thus predict that the observed pattern is driven by a Listener-side bias.

### 3.2.1 Untrained Speaker behavior

For each $i$ drawn from the power-law distribution without replacement, we get a message $m$ from 90 distinct *untrained* Speakers (30 speakers for each hidden size in $[100, 250, 500]$). We experiment with 2 different association processes. In the first, we associate the first generated $m$ to $i$, irrespective of whether it was already associated to another input. In the second, we keep generating a $m$ for $i$ until we get a message that was not already associated to a distinct input. The second version is closer to the MT process (see Section 2.4.2). Moreover, message uniqueness is a reasonable constraint, since, in order to succeed, Speakers need first of all to keep messages denoting different inputs apart.

Figure 2 shows that untrained Speakers have no prior toward outputting long sequences of symbols. Precisely, from Figure 2 we see that the untrained Speakers' average message length coincides with the one produced by the random process defined in Eq. 3 where $p = \frac{1}{a}$.[4] In other words, untrained Speakers are equivalent to a random generator with uniform probability over symbols.[5] Consequently, when imposing message uniqueness, non-trained Speakers become identical to MT. Hence, Speakers faced with the task of producing distinct messages for the inputs, if vocabulary size is not too large, would naturally produce a ZLA-obeying distribution, that is radically altered in joint Speaker-Listener training.

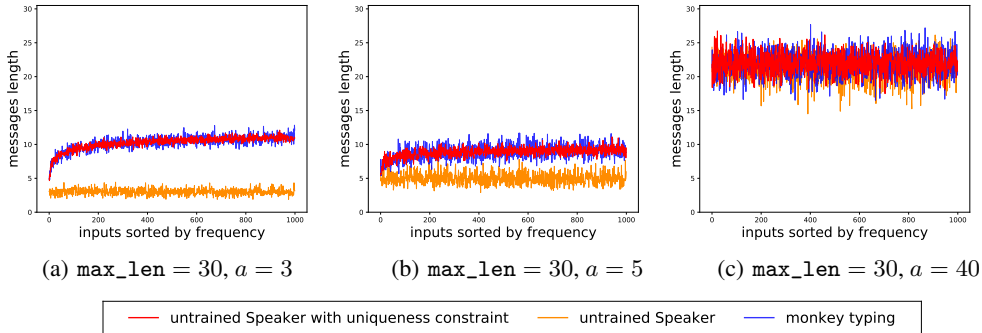

(a) `max_len = 30, a = 3`    (b) `max_len = 30, a = 5`    (c) `max_len = 30, a = 40`

— untrained Speaker with uniqueness constraint    — untrained Speaker    — monkey typing

Figure 2: Average length of messages by input frequency rank for untrained Speakers, compared to MT. See Appendix A.7 for more settings.

### 3.2.2 Untrained Listener behavior

Having shown that untrained Speakers do not favor long messages, we ask next if the emergent anti-efficient language is easier to discriminate by untrained Listeners than other encodings. To this end, we compute the average pairwise L2 distance of the hidden representations produced by untrained Listeners in response to messages associated to all inputs.[6] Messages that are further apart in the representational space of the untrained Listener should be easier to discriminate. Thus, if Speaker associates such messages to the inputs, it will be easier for Listener to distinguish them.

Specifically, we use 50 distinct untrained Listeners with 100-dimensional hidden size.[7] We test 4 different encodings: (1) emergent messages (produced by *trained* Speakers) (2) MT messages (25 runs) (3) OC messages and (4) human languages. Note that MT is equivalent to untrained Speaker, as their messages share the same length *and* alphabet distribution (see Section 3.2.1). We study Listeners' biases with `max_len` = 30 while varying $a$ as messages are more distinct from reference distributions in that case (see Figure A.3 in Appendix A.4). Results are reported in Figure 3. Representations produced in response to the emergent messages have the highest average distance. MT only approximates the emergent language for $a = 1000$, where, as seen in Figure 1 above, MT is anti-efficient. The trained Speaker messages are hence *a priori* easier for non-trained Listeners. The length of these messages could thus be explained by an intrinsic Listener's bias, as conjectured above. Also, interestingly, natural languages are not easy to process by Listeners. This suggests that the emergence of "natural" languages in LSTM agents is unlikely, without imposing *ad-hoc* pressures.

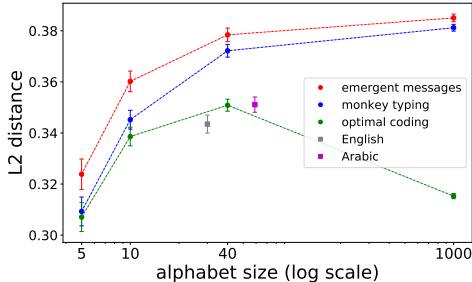

Figure 3: Average pairwise distance between messages' representation in Listener's hidden space, across all considered non-trained Listeners. Vertical lines mark standard deviations across Listeners.

### 3.2.3 Adding a length minimization pressure

We next impose an artificial pressure on Speaker to produce short messages, to counterbalance Listener's preference for longer ones. Specifically, we add a regularizer disfavoring longer messages to the original loss:

$$\mathcal{L}'(i, L(m), m) = \mathcal{L}(i, L(m)) + \alpha \times |m| \qquad (4)$$

where $\mathcal{L}(i, L(m))$ is the cross-entropy loss used before, $|.|$ denotes length, and $\alpha$ is a hyperparameter. The non-differentiable term $\alpha \times |m|$ is handled seamlessly as it only depends on Speaker's parameters $\boldsymbol{\theta}_s$ (which specify the distribution of the messages $m$), and the gradient of the loss w.r.t. $\boldsymbol{\theta}_s$ is estimated via a REINFORCE-like term (Eq. 1). Figure 4 shows emergent message length distribution under this objective, comparing it to other reference distributions in the most human-language-like setting: (`max_len`=30, $a$=40). The same pattern is observed elsewhere (see Appendix A.8, that also evaluates the impact of the $\alpha$ hyperparameter). The emergent messages clearly follow ZLA. Speaker now assigns messages of ascending length to the 40 most frequent inputs. For the remaining ones, it chooses messages with relatively similar, but notably shorter, lengths (always much shorter than MT messages). Still, the encoding is not as efficient as the one observed in natural language (and OC). Also, when adding length regularization, we noted a slower convergence, with a smaller number of successful runs, that further diminishes when $\alpha$ increases.

### 3.3 Symbol distributions in the emergent code

We conclude with a high-level look at what the long emergent messages are made of. Specifically, we inspect symbol unigram and bigram frequency distributions in the messages produced by trained Sender in response to the $1K$ inputs (the `eos` symbol is excluded from counts). For direct comparability with natural language, we report results in the (`max_len`=30, $a$=40) setting, but the patterns are general. We observe in Figure 5(a) that, even if at initialization Speaker starts with a uniform distribution over its alphabet (not shown here), by end of training it has converged to a very skewed one. Natural languages follow a similar trend, but their distributions are not nearly as skewed (see

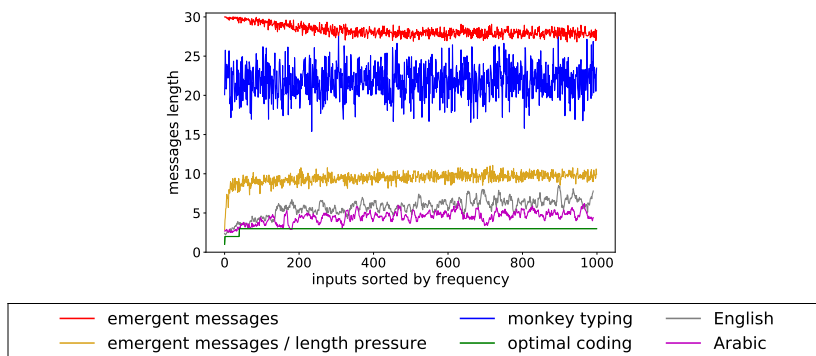

Figure 4: Mean length of messages across successful runs as a function of input frequency rank for `max_len` = 30, $a = 40$, $\alpha = 0.5$. Natural language distributions are smoothed as in Fig. 1.

Figure 8(a) in Appendix A.10 for entropy analysis). We then investigate message structure by looking at symbol bigram distribution. To this end, we build 25 randomly generated *control codes*, constrained to have the same mean length and unigram symbol distribution as the emergent code. Intriguingly, we observe in Figure 5(b) a significantly more skewed emergent bigram distribution, compared to the controls. This suggests that, despite the lack of phonetic pressures, Speaker is respecting "phonotactic" constraints that are even sharper than those reflected in the natural language bigram distributions (see Figure 8(b) in Appendix A.10 for entropy analysis). In other words, the emergent messages are clearly not built out of random unigram combinations. Looking at the pattern more closely, we find the skewed bigram distribution to be due to a strong tendency to repeat the same character over and over, well beyond what is expected given the unigram symbol skew (see typical message examples in Appendix A.9). More quantitatively, across all runs with `max_len`=30, if we denote the 10 most probable symbols with $s_1, ..., s_{10}$, then we observe $P(s_r, s_r) > P(s_r)^2$ with $r \in [\![1, .., 10]\!]$, in more than $97.5\%$ runs. We leave a better understanding of the causes and implications of these distributions to future work.

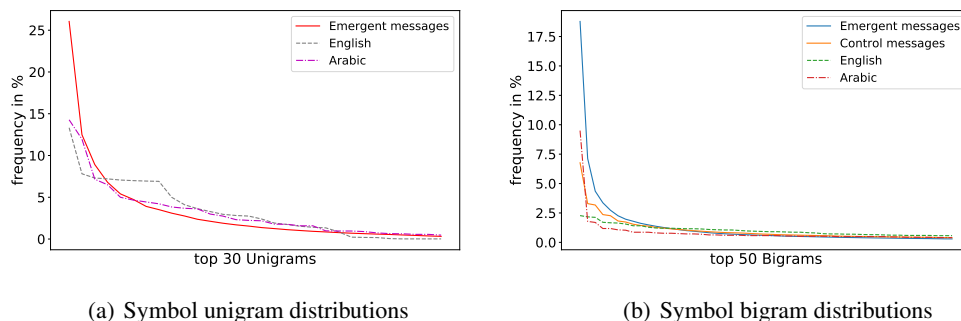

(a) Symbol unigram distributions

(b) Symbol bigram distributions

Figure 5: Distribution of top symbol unigrams and bigrams (ordered by frequency) in different codes. Emergent and control messages are averaged across successful runs and different simulations respectively in the (`max_len`=30,$a$=40) setting.

## 4   Discussion

We found that two neural networks faced with a simple communication task, in which they have to learn to generate messages to refer to a set of distinct inputs that are sampled according to a power-law distribution, produce an *anti-efficient* code where more frequent inputs are significantly associated to longer messages, and all messages are close to the allowed maximum length threshold. The results are

stable across network and task hyperparameters (although we leave it to further work to replicate the finding with different network architectures, such as transformers or CNNs). Follow-up experiments suggest that the emergent pattern stems from an *a priori* preference of the listener network for longer, more discriminable messages, which is not counterbalanced by a need to minimize articulatory effort on the side of the speaker. Indeed, when an artificial penalty against longer messages is imposed on the latter, we see a ZLA distribution emerging in the networks' communication code.

From the point of view of AI, our results stress the importance of controlled analyses of language emergence. Specifically, if we want to develop artificial agents that naturally communicate with humans, we want to ensure that we are aware of, and counteract, their unnatural biases, such as the one we uncovered here in favor of anti-efficient encoding. We presented a proof-of-concept example of how to get rid of this specific bias by directly penalizing long messages in the cost function, but future work should look into less *ad hoc* ways to condition the networks' language. Getting the encoding right seems particularly important, as efficient encoding has been observed to interact in subtle ways with other important properties of human language, such as regularity and compositionality [Kirby, 2001]. We also emphasize the importance of using power-law input distributions when studying language emergence, as the latter are a universal property of human language [Zipf, 1949, Baayen, 2001] largely ignored in previous simulations, that assume uniform input distributions.

ZLA is observed in all studied human languages. As mentioned above, some animal communication systems violate it [Heesen et al., 2019], but such systems are 1) limited in their expressivity; and 2) do not display a significantly *anti*-efficient pattern. We complemented this earlier comparative research with an investigation of emergent language among artificial agents that need to signal a large number of different inputs. We found that the agents develop a successful communication system that does *not* exhibit ZLA, and is actually significantly anti-efficient. We connected this to an asymmetry in speaker vs. listener biases. This in turn suggests that ZLA in communication in general does not emerge from trivial statistical properties, but from a delicate balance of speaker and listener pressures. Future work should investigate emergent distributions in a wider range of artificial agents and environments, trying to understand which factors are determining them.

## 5  Acknowledgments

We would like to thank Fermín Moscoso del Prado Martín, Ramon Ferrer i Cancho, Serge Sharoff, the audience at REPL4NLP 2019 and the anonymous reviewers for helpful comments and suggestions.

## Footnotes

[1]There is always only one message of length 1 (that is, `eos`), irrespective of alphabet size.

[2]No actual monkey was harmed in the definition of the process.

[3]Natural languages have no rigid upper bound on length, and 30 is the highest `max_len` we were able to train models for. Qualitative inspection of the respective corpora suggest that 30 is anyway a reasonable "soft" upper bound on word length in the languages we studied (longer strings are mostly typographic detritus).

[4]Note that we did not use the uniqueness-of-messages constraint to define $P_l$.

[5]We verified that indeed untrained Speakers have uniform probability over the different symbols.

[6]Results are similar if looking at the softmax layer instead.

[7]We fix this value because, unlike for Speaker, it has considerable impact on performance, with 100 being the preferred setting.

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
