[Supplementary Material · zipfian_neurips_2019_supplementary.pdf]

# Anti-efficient message encoding in emergent communication: Supplementary materials

Rahma Chaabouni[1,2], Eugene Kharitonov[1], Emmanuel Dupoux[1,2] and Marco Baroni[1,3]

[1]*Facebook AI Research*
[2]*Cognitive Machine Learning (ENS - EHESS - PSL Research University - CNRS - INRIA)*
[3]*ICREA*
{rchaabouni,kharitonov,dpx,mbaroni}@fb.com

## A.1 Hyperparameters

Both speaker and listener agents are single-layer LSTMs [Hochreiter and Schmidhuber, 1997]. We experiment with the combinations (Speaker's hidden size, Listener's hidden size) in $[(100, 100), (250, 100), (250, 250), (500, 250)]$. We only experiment with combinations where Speaker's hidden-size is bigger or equal to Listener's, because of the asymmetry in their tasks. Indeed, as discussed in Section 3.1 of the main paper, the Speaker's search space $M_a^{\texttt{max\_len}}$ is generally larger than the one of the Listener $R$.

We use the Adam optimizer, with learning rate $0.001$. We apply entropy regularization to Speaker's optimization. The values of the regularization's coefficient are chosen in $[1, 1.5, 2]$. We run the simulation with each hyperparameter setting $4$ times with different random seeds.

## A.2 Monkey typing

We adapt the Monkey typing (MT) process by adding the `max_len` constraint. This makes it a ZLA-like distribution only when vocabulary size $a$ is small. Figure A.1 illustrates this behavior. We see that the higher $a$ is, the further the MT distribution departs from a ZLA pattern.

## A.3 Natural language distributions

We report in Figure A.2 word length distributions for all the natural languages we considered, and compare them with (1) optimal encoding (OC) and (2) emergent language in the most comparable simulation setting: $(\texttt{max\_len} = 30, a = 40)$. Despite their different alphabet sizes, natural languages pattern similarly: They follow ZLA, and approximate OC.

## A.4 Anti-efficient emergent language

Figure A.3 shows message length distribution (averaged across all successful runs) in function of input frequency rank, and compares it with some reference distributions. The results are in line with our finding in Section 3.1 of the main paper.

## A.5 Emergent language with uniform input distribution

Agents' messages are very long also when the input distribution is uniform, see Figure A.4. Their average length is significantly larger than MT messages with uniform inputs (t-test, $p < 10^{-9}$).

(a) `max_len`=6, $a$=5     (b)`max_len`=6, $a$=10     (c)`max_len`=6, $a$=40

(d) `max_len`=11, $a$=3    (e) `max_len`=11, $a$=5    (f)`max_len`=11, $a$=10    (g)`max_len`=11, $a$=40

(h)`max_len`=30, $a$=3    (i)`max_len`=30, $a$=5    (j)`max_len`=30, $a$=10    (k)`max_len`=30, $a$=40

Figure A.1: Monkey typing encoding: Mean message length across 50 simulations as a function of input frequency rank.

Figure A.2: Word length in natural languages in function of word frequency rank, compared to average emergent code and OC in the ($\mathtt{max\_len} = 30, a = 40$) setting. For readability, we smooth natural language distributions by reporting the sliding average of 10 consecutive lengths.

## A.6 Randomization test

In the main paper, we observe a tendency for Speaker to use longer messages for frequent inputs, making its code obey a sort of "*anti*-ZLA". In this section, we provide quantitative support for this observation. We run the randomization test of Ferrer i Cancho et al. [2013]. We note $E = \sum_{i=1}^{1000} p_i \times l_i$ the mean length of messages, where $p_i$ is the probability of the type $i$ and $l_i$ is the length of the corresponding message. A language that respects ZLA is characterized by a small $E$ (optimal coding, OC, is associated with $min(E)$). Under $H_0$, the mean length of the encoding coincides with the mean length of a random permutation of messages across types. To be comparable with Ferrer i Cancho et al. [2013], we use the same number of permutations ($= 10^5$). Also, we adopt

Figure A.3: Mean message length across successful runs as a function of input frequency rank, with reference distributions. Natural language distributions are smoothed as in Fig. A.2.

their definition of "left p-value" and "right p-value". If left p-value$\leq 0.005$, the studied encoding is *significantly small* (characterized by significantly smaller $E$ than random permutations), if right p-value$\leq 0.005$, it is *significantly large*, corresponding to our notion of anti-efficiency.

We observe in Table A.1 that $H_0$ is only rejected for MT with $a \geq 40$, which, as we mentioned in the main paper, approaches a random length distribution for those cases, and for emergent messages with $a = 1000$. OC, natural languages, and emergent language *with* Speaker-length regularization are, in all the considered settings, significantly more efficient than chance. Importantly, the Emergent language results confirm LSTMs' natural preference for long messages ($E$ approaching `max_len`) and *significant* anti-efficiency for $a \leq 40$ (right p-value$\approx 0$). When $a = 1000$, there is no frequency rank/length relation and all lengths $\approx$ `max_len`.

## A.7 Speaker initial length distribution

Figure A.5 plots message length in function of input frequency rank for several settings. In particular, we report *all* settings $(\texttt{max\_len}, a)$ that succeeded when training the Speaker-Listener system. Here,

(a) $a=3$     (b) $a=5$     (c) $a=10$     (d) $a=40$

— Emergent messages     — Monkey typing     — Optimal coding

Figure A.4: Mean message length per input across successful runs for `max_len=30` and different $a$. Inputs are uniformly distributed.

Table A.1: Results of the randomization test for different codes when $\texttt{max\_len} = 30$ and with different alphabet sizes $a$. Left/right p-values significant at $\alpha = 0.01$ suffixed by asterisk. See Table 1 of Ferrer i Cancho et al. [2013] for more codes to be compared with our results.

| Setting | Code | $E$ | Left p-Value | Right p-Value |
|---|---|---|---|---|
| $a = 5$ | OC | 3.55 | $< 10^{-5}*$ | $> 1 - 10^{-5}$ |
| | MT | 7.56 | $< 10^{-5}*$ | $> 1 - 10^{-5}$ |
| | Emergent | 26.98 | $> 1 - 10^{-5}$ | $< 10^{-5}*$ |
| $a = 10$ | OC | 2.82 | $< 10^{-5}*$ | $> 1 - 10^{-5}$ |
| | MT | 11.27 | $0.0002*$ | 0.998 |
| | Emergent | 26.73 | $> 1 - 10^{-5}$ | $< 10^{-5}*$ |
| $a = 40$ | OC | 2.29 | $< 10^{-5}*$ | $> 1 - 10^{-5}$ |
| | MT | 21.30 | 0.814 | 0.186 |
| | Emergent | 29.40 | $> 1 - 10^{-5}$ | $< 10^{-5}*$ |
| | Regularized ($\alpha$=0.5) | 7.22 | $< 10^{-5}*$ | $> 1 - 10^{-5}$ |
| | English | 3.68 | $< 10^{-5}*$ | $> 1 - 10^{-5}$ |
| | Arabic | 3.14 | $< 10^{-5}*$ | $> 1 - 10^{-5}$ |
| $a = 1000$ | OC | 1.86 | $0.001*$ | 0.999 |
| | MT | 29.67 | 0.750 | 0.250 |
| | Emergent | 29.98 | 0.072 | 0.928 |

however, no training is performed, so that we can observe Speaker's initial biases. The results are in line with our finding in Section 3.2.1 of the main paper.

## A.8 The effect of length regularization

We look here at the effect of the regularization coefficient $\alpha$ on the nature of the emergent encoding. To this end, we consider the setting that is least efficient when no optimization is applied: ($\texttt{max\_len} = 30, a = 1000$). The same pattern is also observed with different choices of `max_len` and $a$. Figure A.6 shows, for $\alpha = 1$, that emergent messages *approximate optimal coding*. For even larger values, we were not able to successfully train the system to communicate. This is in line with Zipf's view of *competing* pressures for accurate communication vs. efficiency. The emergent messages follow ZLA only when both pressures are at work. If the efficiency pressure is not present, agents come up with a communicatively effective but non-efficient encoding, as shown in Section A.4 and Section 3.1 of the main paper. However, if the efficiency pressure is too high, agents cannot converge on a protocol that is successful from the point of view of communication.

Figure A.5: Average length of messages in function of input frequency rank for untrained Speakers, compared to MT. In each figure we report the results in a specific setting $(\texttt{max\_len}, a)$.

Figure A.6: Length of messages as a function of input frequency for $\texttt{max\_len} = 30$ and $a = 1000$, when varying $\alpha$ in the length regularization case.

## A.9 Repetition in emergent messages

We report in listings 1, 2, 3 and 4 examples of emergent messages in different settings. We notice that the agents extensively use repetition, even when $a$ (vocabulary size) is large. This repetition, that results in the very skewed bigram distributions presented in Section 3.3 of the main paper, increases with higher $\texttt{max\_len}$, as shown in figure A.7. Moreover, from figure A.7, we see that, unlike in emergent codes, this sort of repetition does not appear in natural language.

Listing 1: Emergent messages for the 4 most frequent inputs ($\texttt{max\_len}$:11 and $a$:40).

```
m1:  18,5,36,36,5,5,10,5,32,8,eos
m2:  1,36,2,36,10,13,9,29,33,eos
m3:  29,1,8,1,39,39,9,15,10,19,eos
```

```
m4:  29 ,1 ,36 ,36 ,36 ,36 ,5 ,8 ,13 ,9 , eos
```

Listing 2: Emergent messages for the $4$ most frequent inputs (`max_len`:11 and $a$:1000).

```
m1:  431 ,431 ,305 ,305 ,70 ,70 ,331 ,391 ,134 ,581 , eos
m2:  867 ,288 ,466 ,466 ,466 ,737 ,113 ,77 ,615 ,615 , eos
m3:  288 ,466 ,466 ,466 ,418 ,144 ,113 ,615 ,638 ,615 ,  eos
m4:  4 ,4 ,152 ,152 ,152 ,468 ,642 ,615 ,422 ,134 , eos
```

Listing 3: Emergent messages for the $4$ most frequent inputs (`max_len`:30 and $a$:5).

```
m1:  3 ,4 ,4 ,4 ,1 ,1 ,1 ,1 ,1 ,1 ,1 ,1 ,1 ,4 ,4 ,4 ,4 ,4 ,4 ,4 ,4 ,4 ,4 ,4 ,4 ,3 ,4 ,3 ,4 , eos
m2:  3 ,1 ,3 ,3 ,1 ,1 ,1 ,1 ,1 ,1 ,1 ,4 ,4 ,4 ,4 ,4 ,2 ,4 ,2 ,4 ,2 ,4 ,2 ,4 ,2 ,4 ,2 ,4 ,3 ,2 , eos
m3:  1 ,4 ,4 ,1 ,1 ,1 ,1 ,1 ,1 ,1 ,1 ,4 ,4 ,4 ,4 ,4 ,2 ,4 ,4 ,4 ,4 ,4 ,4 ,4 ,4 ,4 ,2 ,4 ,3 ,1 , eos
m4:  1 ,4 ,4 ,1 ,1 ,1 ,1 ,1 ,1 ,1 ,4 ,4 ,4 ,4 ,4 ,4 ,4 ,4 ,4 ,4 ,4 ,4 ,2 ,4 ,2 ,2 ,4 ,1 ,4 , eos
```

Listing 4: Emergent messages for the $4$ most frequent inputs (`max_len`:30 and $a$:40).

```
m1:  11 ,11 ,12 ,24 ,8 ,8 ,12 ,24 ,12 ,12 ,12 ,12 ,12 ,12 ,36 ,24 ,24 ,35 ,35 ,35 ,36 ,36 ,20 ,15 ,36 ,19 ,11 ,31 ,13 ,  eos
m2:  13 ,31 ,31 ,24 ,8 ,8 ,8 ,8 ,8 ,8 ,8 ,8 ,19 ,24 ,3 ,3 ,36 ,36 ,19 ,29 ,15 ,31 ,30 ,31 ,15 ,19 ,11 ,13 , eos
m3:  39 ,8 ,12 ,8 ,8 ,8 ,8 ,25 ,25 ,25 ,25 ,25 ,25 ,25 ,36 ,24 ,12 ,12 ,35 ,35 ,35 ,18 ,18 ,11 ,3 ,7 ,11 ,7 ,11 , eos
m4:  14 ,31 ,8 ,8 ,8 ,8 ,8 ,24 ,25 ,25 ,25 ,36 ,36 ,36 ,36 ,36 ,36 ,36 ,36 ,36 ,3 ,2 ,35 ,30 ,31 ,21 ,29 , eos
```

Figure A.7: Mean message length (weighted by input probability, and averaged across successful runs) for various `max_len` and fixed $a = 40$, before and after removing all repetitions. A repetition here refers to a sequence of 2 or more consecutive identical symbols. Emergent messages are indexed by their `max_len`, and we add the same statistics in two human languages for comparison.

## A.10   Entropy of symbol distributions in different codes

We report the entropy of symbol unigram and bigram distributions for different codes in figures 8(a) and 8(b), respectively. We observe that, in both cases, the emergent code symbol distribution is more skewed than in any considered reference code.

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

(a) Entropy of unigram distributions

(b) Entropy of bigram distributions

Figure A.8: Entropy of symbol unigram and bigram distributions for different codes (in natural log). The higher the entropy, the more uniform the corresponding distribution is. The entropy of the uniform code is computed by assuming a uniform distribution over 40 symbols (unigram) and 1600 sequences of 2 symbols (bigram). MT and control messages (see Section 3.3 of main text) are averaged across 25 different simulations in the (max_len=30,$a$=40) setting. Emergent messages are averaged across successful runs in the same setting.