[Reviews · NeurIPS 2019]

Reviewer 1



**Update after author response** The response addresses my main concerns from the review and I still recommend acceptance. === This paper provides a focused study of the distribution of message lengths in an emergent communication task. A Lewis-type signaling game is constructed in which referents are generated from a power-law distribution. RNN "speaker" and "listener" models are constructed to communicate via a discrete channel (with variable vocabulary size and max length) and trained to maximize success at the signaling game using a vanilla policy gradient algorithm. It is observed that more frequent referents are associated with *longer* messages from the speaker agent. This is in contrast to natural language (exemplified by corpus data from English and Arabic and two simple computational models). Further studies show that this "unnatural" length distribution can be corrected by adding a simple penalty, and suggest that it is driven by initial conditions for both the speaker and listener models. STRENGTHS - thorough empirical study relevant to ongoing work on emergent communication - nice literature review and well-motivated reference models WEAKNESSES - some experimental & analysis details are unclear or omitted - corpus baselines (and the status of "referents" in natural language generally) are not super well aligned with the main task in the paper While I have a few questions about the experimental setup and some of the linguistic claims, I think this is a thorough and well-executed empirical paper and a useful contribution to the emergent communication literature. I think it should be accepted. REFERENT DISTRIBUTIONS This is my main substantive complaint: line 43 says "[referents are] randomly drawn from a power-law distribution (so that referent frequency is extremely skewed, as in natural language)". This conflates the distribution of *words* with the distribution of *referents*. ZLA says nothing about the distribution of referents. A claim that referents in natural language are also distributed as a power law is non-trivial and requires citation, given that most words (especially the frequent ones!) have no referential function on their own. For the same reason, I'm a little uncomfortable with the models of natural language built from corpus word frequency data, since the processes determining word frequencies in these corpora are fundamentally different from those producing the agent behavior in this paper. I don't think any of this changes the bottom line, and you should definitely keep all the current experiments, but I would like to see this paper be a little more careful about the difference between words, referents, and their associated frequencies in both the motivation section and in comparisons to real-world language data. EXPERIMENTS AND ANALYSIS The central claim in this paper is that message length is anticorrelated with frequency in the base model, but this is only backed up visually (i.e. "the line goes down in Figure 1"). We really need to see a correlation coefficient and a hypothesis test somewhere.... There are a couple of cases where numbers get averaged across training runs, and I'm unclear about what's being averaged. In Fig 2, are we looking at the average length of all rank-i messages? Or the average rank of all length-l messages? Similarly, in Fig 3, are we looking at (average pairwise distance) averaged across training runs? Or (average distance across training runs) averaged pairwise? There are a couple of empirical claims for which no quantitative statement (or even appendix pointer) is provided: "The higher D is, the more accurate the agents become" on 179; "the patterns are general" on 256. MESSAGE DISTRIBUTIONS 31: "There is an inverse (non-linear) correlation between word frequency and length". The precise nature of this non-linearity is one of the most interesting underlying computational-linguistic issues here, and I think the paper would benefit from exploring it a bit more. Of the citations provided for this line, one argues for a power-law distribution of word frequencies, and one argues for a Gamma distribution. The "optimal code" model in this paper has an exponential word frequency distribution, and I think monkey typing probably does as well. What about the learned agents? In addition to plotting things on an absolute message length / frequency scale, it would be extremely interesting to normalize them in some way and try to make a claim about the functional form of the length distribution (esp. as it appears in the penalized vs unpenalized experiment) with an appropriate statistical test (K-S etc.). As a related presentation issue, I think all of the figures in this paper would be clearer on log or log-log scales. MISCELLANEOUS - 15: s/strand/stray? - 21: Nitpick---I find this use of "AIs" needlessly imprecise. Perhaps "automated agents" or "computational agents" instead? - 57: cite Lewis 69 "Convention" for the signaling game. - Eq 1 is Williams 92, right? What's been added from Schulman 15? - 157: "xdistributions" - What does distribution of lengths look like w/r/t a uniform distribution over referents?

Reviewer 2



After author response, I am keeping my score the same. The authors have promised to add the most important detail I felt was lacking: whether a cost to communication lead to reduced communication success. ----------- This paper investigates emergent communication between two agents in a very simple communication game where the referents are power-law distributed. The authors show that naively training the agents with reinforcement learning using policy gradient methods lead to communication protocols where common referents are associated with longer messages, which contradicts Zipf's Law. This is an important observation which serves to delineate artificial agent communication from human communication. The authors then provide an elegant and simple explanation for why this might be the case grounded in the representational capacity of the listener agent, and Figure 3 seems to provide good evidence for their explanation. It is interesting to speculate whether this is caused by a peculiarity in LSTM dynamics, and whether encoders with alternative architectures (such as hierarchical tree-based encoders) distinguish different features. Further, the authors show that a simple length penalty eliminates the anti-efficient coding behaviour, and results in communication which exhibits a Zipfian distribution. This shows that the conventional view that communication is costless may not be entirely accurate. The authors do not state whether the length penalty affects communication success; this would serve as an interesting comparison. Finally, the authors investigate the symbol statistics of the resulting communication protocol. This is (for me) the weakest section of the paper, as I do not feel it contributes much to the main thrust of the argument. However, the other two sections by themselves justify acceptance. The discussion points raised by the authors are all worthy of further thought. Overall, I feel that this paper raises an interesting and important observation. Do we care solely about agents communicating with each other to achieve task success in situations where success is not possible without communication, or do we study emergent communication as a tool to study the evolution of human language? If the latter, then the exact way we define tasks and train our agents can provide important clues about what pressures shaped human language, and this paper proposes and investigates one such pressure: an inherent cost to producing sounds. The paper is clearly written, and deserves a wider audience at NeurIPS.

Reviewer 3



The paper studies properties of a language that emerges by solving communication task with a power-law distributed referents. The direction of studying implicit preferences of the recurrent neural networks from the perspective of emergent language is quite important and understanding such properties can improve existing models. The paper is well structured and contains a very detailed description of all performed experiments.

[Author Response · NeurIPS 2019]

Thanks for the very constructive feedback. Due to lack of space, we only address here the major issues that were raised.
We will however incorporate all feedback in our paper revision.

**Power-law distribution of input referents (R1/R3).** We agree with the reviewers that our assumption that words
in natural language are power law-distributed because their referents in the world are is unwarranted. A more careful
characterization for our setup is that the inputs to the Speaker represent abstract word types (which are definitely power
law-distributed in languages); the task of the Speaker agent is to map these abstract types to phonological/orthographic
forms and vice versa for the Listener agent. This brings our setup closer to the case of natural language; we will
rephrase this in the introduction and discussion accordingly.

**Uniform input distribution (R1/R3).** Agents' messages are very long also when the input distribution is uniform,
see Fig 1 (to be included in Supplementary with more settings, that follow the same pattern). Their average length is
significantly larger than MT messages with uniform inputs (t-test, $p < 10^{-9}$).

**Quantitative support for "anti-efficiency" claim (R1).** Instead of running correlations which make assumptions
about the underlying distribution, we have run the randomization test of Ferrer-i-Cancho et al. (CogSciJ 2013). We
note $E = \sum_{i=1}^{1000} p_i \times l_i$ the mean length of messages, where $p_i$ is the probability of the type $i$ and $l_i$ is the length
of the corresponding message. A language that respects ZLA is characterized by a small $E$ (optimal coding, OC, is
associated with $min(E)$). Under $H_0$, the mean length of the encoding coincides with the mean length of a random
permutation of messages across types. Also, we adopt Ferrer-i-Cancho et al. (CogSciJ 2013) definition of "left
p-value" and "right p-value". If left p-value$\leq 0.005$, the studied encoding is *significantly small* (characterized by
significantly smaller $E$ than random permutations), if right p-value$\leq 0.005$, it is *significantly large*, corresponding to
our notion of anti-efficiency. We observe in Table 1 (to be included in Supplementary with more settings, that confirm
the same pattern) that $H_0$ is not rejected only for MT, which, as we mentioned in the paper, approaches a random
length distribution for large $a$. OC, natural languages, and emergent language *with* Speaker-length regularization are
significantly more efficient than chance. Importantly, the Emergent language results confirm LSTMs' natural preference
for long messages ($E$ approaching `max_len`) and *significant* anti-efficiency (right p-value$\approx 0$).

| code | $E$ | left p-value | right p-value |
|---|---|---|---|
| OC | 2.29 | $< 0.005$ | 1 |
| MT | 21.30 | 0.81 | 0.18 |
| Emergent | 29.40 | 1 | $< 0.005$ |
| Regularized ($\alpha$=0.5) | 7.22 | $< 0.005$ | 1 |
| English | 3.68 | $< 0.005$ | 1 |
| Arabic | 3.14 | $< 0.005$ | 1 |

Figure 1: Mean message length per word type across successful runs, `max_len`=30, $a$=40. Word types are uniformly distributed.

Table 1: Randomization test results for `max_len`=30, $a$=40. OC: Optimal Coding, MT: Monkey Typing. To be comparable with previous studies, we use the same parameters as in Ferrer-i-Cancho et al. (CogSciJ 2013).

**Specific points**
*R1: There are a couple of cases where numbers get averaged [...], and I'm unclear about what's being averaged.*
Figure 2: average length of all rank-i messages across successful runs. Figure 3: average pairwise distance across all
considered non-trained Listeners. We will clarify accordingly in the paper.
*R2: It is interesting to speculate whether this is caused by a peculiarity in LSTM dynamics, and whether encoders with*
*alternative architectures (such as hierarchical tree-based encoders) distinguish different features.*
Very interesting idea; we have indeed preliminary results suggesting that a Transformer listener may be less anti-efficient
than LSTM. To be further explored in future work.
*R2: The authors do not state whether the length penalty affects communication success.*
Convergence is slower with smaller number of successful runs (depending on the coefficient $\alpha$) in this case. We will
report this in the paper.
*R3: Somewhat unsurprisingly, the developed protocols implement "anti- efficient" encoding.*
We were actually surprised by this. Ours is the first successful protocol ever to display a significant anti-efficient effect
(compare to natural languages and animal communication systems in Ferrer-i-Cancho et al CogSciJ 2013).
*R3: The authors mentioned they use top 1000 most frequent words from natural languages. Do they have the same*
*degree (exponent) of a power-law distribution as in synthetic referents experiment?*
The natural languages corpora follow a power-law distribution with an exponent between -0.81 and -0.92 (we used $-1$
in the artificial language).

[Meta-Review · NeurIPS 2019]

The submission presents a carefully designed study of the message length distribution in emergent communication. Though the reviewers disagree on how surprising the finding is (e.g., R3 believes it directly follows from the reward structure), the paper appears uncontroversial, as all the reviewers agree that it will be interesting and important for anyone working on this problem. The author response was crucial in clarifying several points in the paper, and I hope that the authors will update the paper accordingly. I also agree with the reviewers that the assumption about the distribution of reference may be questionable, and hope that the authors will keep their promise and include results with the uniform input distribution.